# Dynamic arm movements attenuate the perceptual distortion of visual vertical induced during prolonged whole-body tilt

**Keisuke Tani**[1]*, **Shinji Yamamoto**[2], **Yasushi Kodaka**[3], **Keisuke Kushiro**[4]

**1** Laboratory of Psychology, Hamamatsu University School of Medicine, Hamamatsu, Shizuoka, Japan, **2** Faculty of Sport Sciences, Nihon Fukushi University, Mihama-cho, Aichi, Japan, **3** Automotive Human Factors Research Center, National Institute of Advanced Industrial Science and Technology, Tsukuba, Ibaraki, Japan, **4** Graduate School of Human and Environmental Studies, Kyoto University, Kyoto, Japan

* keisuketani.pt@gmail.com

**Data Availability Statement:** All individual dataset are available from Open Source Framework (OSF) at https://doi.org/10.17605/OSF.IO/BUFTG.

## Abstract

Concurrent body movements have been shown to enhance the accuracy of spatial judgment, but it remains unclear whether they also contribute to perceptual estimates of gravitational space not involving body movements. To address this, we evaluated the effects of static or dynamic arm movements during prolonged whole-body tilt on the subsequent perceptual estimates of visual or postural vertical. In Experiment 1, participants were asked to continuously perform static or dynamic arm movements during prolonged tilt, and we assessed their effects on the prolonged tilt-induced shifts of subjective visual vertical (SVV) at a tilted position (*during-tilt* session) or near upright (*post-tilt* session). In Experiment 2, we evaluated how static or dynamic arm movements during prolonged tilt subsequently affected the subjective postural vertical (SPV). In Experiment 1, we observed that the SVV was significantly shifted toward the direction of prolonged tilt in both sessions. The SVV shifts decreased when performing dynamic arm movements in the *during-tilt* session, but not in the *post-tilt* session. In Experiment 2, as well as SVV, the SPV was shifted toward the direction of prolonged tilt, but it was not significantly attenuated by the performance of static or dynamic arm movements. The results of the *during-tilt* session suggest that the central nervous system utilizes additional information generated by dynamic body movements for perceptual estimates of visual vertical.

## Introduction

Knowledge of the gravitational direction is fundamental to our action and perception of the earth. The direction of gravity cannot be directly sensed; instead, it is estimated in the brain based on several types of sensory information. Numerous psychophysical studies have demonstrated the involvement of visual [1–3], somatosensory [4–6], and vestibular sensory signals [3, 7] in estimates of gravitational direction. Moreover, recent studies using computational modeling have shown that the central nervous system (CNS) weighs and combines these multisensory signals with prior knowledge and experience about the earth-vertical direction in a statistically optimal manner to resolve sensory ambiguity [7–9]. One typical way to evaluate internal estimates of the gravitational direction is the subjective visual vertical (SVV)

**Funding:** The present study was funded by Japan society for the promotion of science (JSPS) KAKENHI No. 17J07245 and No. 20K19305 (to KT), and No. 16K01595 and No. 19K11621 (to KK). The funder had no role in study design, data collection and analysis, decision to publish, or preparation of the manuscript.

**Competing interests:** The authors have declared that no competing interests exist.

adjustment, in which participants are asked to adjust a visual line to the perceived vertical [10]. Although the SVV closely coincides with the actual gravitational vertical in the upright position, the estimation error occurs when the head or body are tilted [11–14]. For instance, for a relatively small tilt angle ($< 60°$), the SVV typically shifts toward the opposite direction of body tilt [15]. Another method of assessing the perception of the gravitational direction is the subjective postural vertical (SPV) task, in which participants are asked to indicate their body's vertical position while being inclined from one tilted side to the other [16]. It is known that although the estimation of postural vertical is relatively accurate, the SPV angle is affected by the direction and angle of the initial body tilt [16, 17].

The perception of gravitational direction is affected by maintaining the body in an inclined posture for a certain time, referred to as prolonged tilt. The SVV gradually shifts toward the tilted side during prolonged tilt [18–20] and remains deviated toward the previously tilted side even after a return to the upright position (i.e., after-effect) [19, 21–23]. Likewise, after prolonged tilt, the SPV biases toward the direction of prolonged tilt [24–27]. These time-dependent changes in SVV and SPV may be mainly attributable to sensory adaptation. Fernandez and Goldberg [28] showed that the otolith afferent firing rate in primates gradually decreased in the roll head-tilted position. Other studies suggest that somatosensory adaptation derived from trunk receptors may also contribute to the SVV shifts during prolonged tilt [11, 23]. The angles of the head and body relative to gravity would be sensed to be smaller due to vestibular and somatosensory adaptation, leading to shifts of the perceived direction of gravity toward the direction of prolonged tilt [29].

The present study aimed to investigate the effect of active arm movements on the perceptual estimates of gravitational direction. Performing arm movements against gravity generates additional information, such as proprioceptive feedback from muscle spindles, skin and joint receptors, and the Golgi tendon organ, as well as efferent copy [30], which would provide cues about the gravitational force on the arm. Moreover, the gravitational torque on the shoulder of an extended arm during arm lifting depends on the position of the arm relative to gravity [31]. Therefore, the gravitational cues generated by arm movements would play a role in estimating the gravitational direction. Previous studies have shown that the body tilt-induced errors in the judgment of the head-referenced eye level considerably decreased when accompanied by arm movements during judgment [32, 33]. This finding suggests that active body movements can improve the accuracy of spatial judgments, but it is unknown whether active body movements also influence the perceptual estimates of gravitational space not involving body movements. The CNS considers prior knowledge and experience as well as sensory signals to estimate the gravitational vertical [7–9], allowing us to hypothesize that additional cues generated by body movements may contribute to the subsequent perceptual estimates of the gravitational direction via prior knowledge and/or experience. To test this hypothesis, the present study evaluated whether static or dynamic arm movements during prolonged tilt influenced the perceptual judgments of visual vertical (Experiment 1) or postural vertical (Experiment 2). As mentioned above, the internal estimates of the gravitational direction are distorted during or after prolonged tilt, primarily due to sensory adaptation. We expected that these distorted estimates might be corrected based on additional cues generated by arm movements, resulting in the maintenance of SVV or SPV angles even after prolonged tilt.

# Experiment 1

## Materials and methods

**Participants.** Fifteen right-handed healthy volunteers (13 males and 2 females, aged 19–33 years) participated in this experiment after providing written informed consent. All

participants had normal vision and no neurological, muscular, or cognitive disorders. This study was approved by the Ethics Committee of the Graduate School of Human and Environmental Studies, Kyoto University, and was conducted in accordance with the Declaration of Helsinki (2013).

**Apparatus.** The participants sat on a seat (RSR-7 KK100, RECARO Japan, Japan) mounted on a tilt table in a completely dark room. The head, trunk, and legs were firmly secured to the seat with bands and a four-point safety belt in a natural position (Fig 1). An axis under the tilt table was expanded or contracted via a servo motor, enabling the tilt table to be tilted in the roll plane around a rotation center located 18 cm underneath the bottom of the seat. The tilting velocity and initial acceleration were 0.44°/s and 0.09°/s², respectively, which are below the rotational acceleration threshold [34]. Therefore, in the present study, the contribution of the semi-circular canal to the estimation of the visual vertical would be negligible.

A display (LTN097QL01, SAMSUNG, Korea; 19.6 cm × 14.7 cm) was placed 35 cm in front of the participant's face. To prevent any spatial cues such as the edge of the display, a black-colored cylinder (26 cm in diameter) with one end covered by a plate with a hole (10 cm in diameter) in the center was placed between the face and the display. During the SVV adjustment, a white line (length, 4 cm; width, 0.1 cm) that was rotated via a digital controller (BSGP1204, iBUFFALO, Japan) was presented at the center of the display. An anti-aliasing mode was applied to the projection to avoid any orientational cues derived from the pixel alignment. The display was mounted on the tilt table via metal frames (Green Frame, SUS, Japan), maintaining identical display positions relative to the participants regardless of the body tilt angle. The center of a vertical frame positioned on the left side of the tilt table had a hinge structure, enabling the display portion to be rotated in the yaw plane independently of the tilting chair. Before the

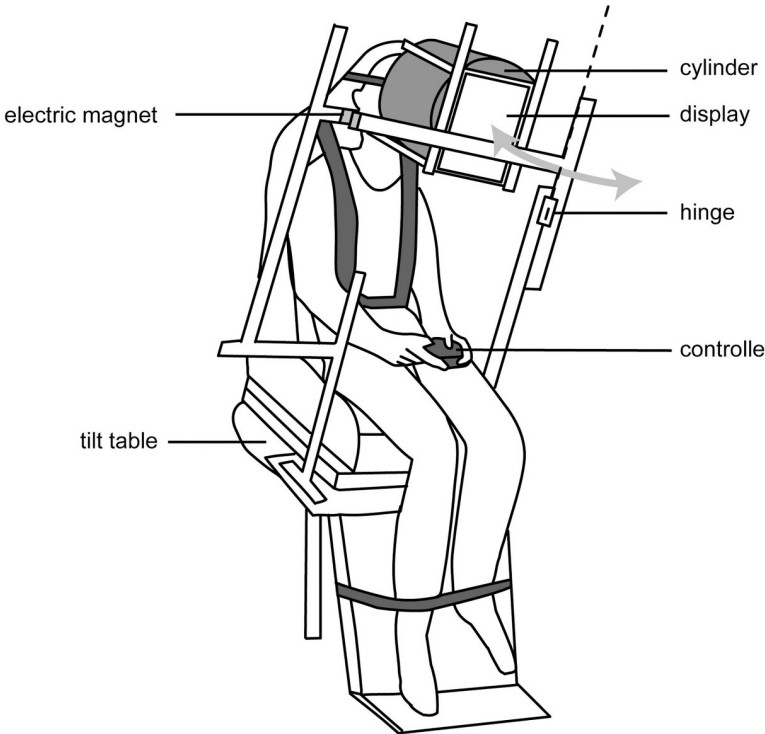

**Fig 1. Schematic illustration of the experimental setup.** This figure illustrates a situation in which the participant was tilted leftward. The display portion was rotated in yaw, as denoted by a gray arrow.

participants performed the task (see *Task during prolonged tilt* in detail), an experimenter rotated the display portion to the left side of the participants, preventing them from hitting their arm against the display or frames. An electric magnet was placed between the display portion and the horizontal frame positioned on the right side of the tilt table. The display portion and frames were firmly fixed via electrification of the electric magnet, enabling it to be set in front of the face. Prior to each experiment, the angle of the tilt table and the upper side of the display relative to the floor was calibrated at 0˚ using a digital inclinometer.

To temporarily restrict vision, the participants wore a mechanical shutter goggle controlled via a microcomputer (Arduino UNO, Arduino SRL, United States) during the experiment. They also wore earphones via which white noise was provided to avoid auditory cues from the environment.

**Experimental procedure.**    This experiment consisted of two sessions: *the during-tilt* and *post-tilt* sessions. In the former, we evaluated how the SVV was influenced by arm movements at the tilted position. In the latter, we confirmed whether the effects of arm movements during prolonged body tilt influenced the SVV after returning to the near-upright positions (0˚ or ±4˚). These angles were determined based on the fact that 4˚ is the threshold for the detection of body tilt in the roll plane [35]. The order of each session was randomized for each participant.

Fig 2A shows a sequence of experimental trials in the *during-tilt* session. After the shutter was closed, the tilt table was tilted to the left. One second after the tilt table came to the left-side-down (LSD) 16˚ position, the shutter opened again, and a white line was presented on the display. The participants were asked to adjust the line to the gravitational vertical via the controller (SVV adjustment). The initial angle of the line was set at ±45˚, ±60˚, or 90˚ relative to the body longitudinal axis in a pseudorandomized order. The participants performed five trials of the SVV adjustment within 40 seconds. The shutter then closed, and the display portion was moved leftward by the experimenter. The participants were asked to execute one of three tasks (see *Task during prolonged tilt*) at the tilted position. After the display portion was returned to the initial position (i.e., in front of the participant's face), the shutter opened and the participants were asked to perform the SVV adjustments for five trials again. Each participant performed this sequence of experimental trials for each task condition, that is, 30 trials (three task conditions [No-movement, Static, Dynamic tasks] × 2 phases [before, after task] × 5 SVV adjustments) in total. A break of approximately 2 min was given between conditions. The order of the task conditions was pseudorandomized for each participant.

Fig 2B shows a sequence of experimental trials in the *post-tilt* session. After the participants were tilted to the LSD 16˚ position from upright with the shutter closed, the display portion was moved leftward by the experimenter, and the participants were asked to perform one of the three tasks during prolonged tilt. The display portion was then moved back to the original position, and the body was tilted to one of three final tilt positions: upright, right-side-down (RSD) 4˚, or LSD 4˚. The shutter was opened, and the participants were asked to repeat the SVV adjustments for five trials. After completing the task, the body was returned to the upright position via the RSD 16˚ position to avoid providing feedback about the final tilt position that could influence the subsequent performance on the SVV adjustment. In this session, the participants were not informed of the angles of the final tilt positions. Each participant performed this sequence of trials for each task condition in each final tilt position, that is, 45 trials (three task conditions [No-movement, Static, Dynamic tasks] × 3 final tilt positions [0˚, ±4˚] × 5 SVV adjustments) in total.

After all trials were completed, participants performed the SVV adjustments for five trials in each final tilt position (0˚ and ±4˚) immediately after being tilted from an upright position (not via the LSD position), referred to as the Control condition (Fig 2C). Note that the effect of

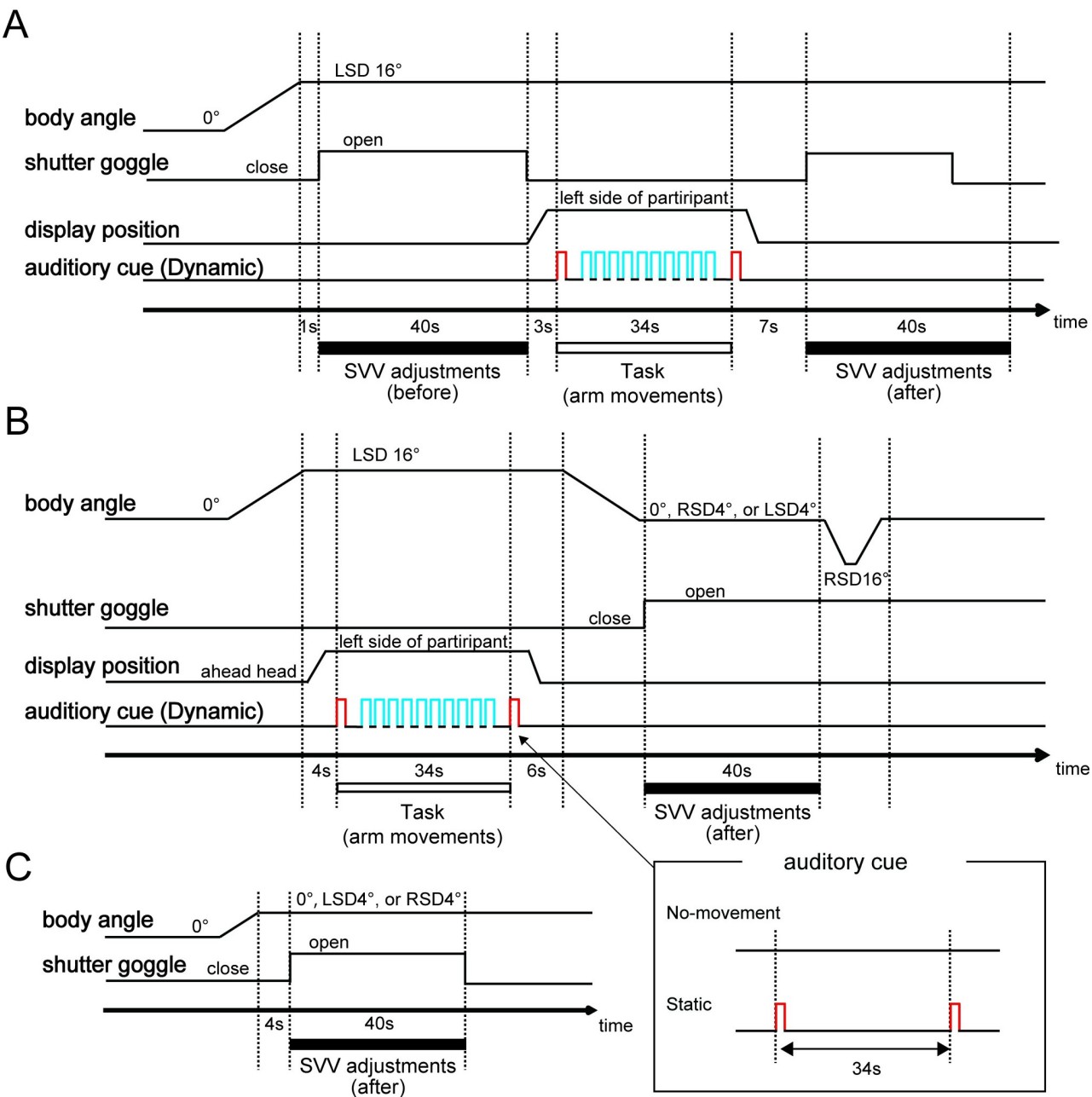

**Fig 2.** Schematic illustration of the experimental procedure in the *during-tilt* (A) and *post-tilt* sessions (B) and the Control condition (C). In both sessions, participants were asked to perform one of three tasks (No-movement, Static, or Dynamic task) during prolonged tilt in response to preparation (denoted as *red*) or action sounds (denoted as *blue*). As an example, the auditory cue for the dynamic task is shown in figures (A) and (B).

prolonged tilt (including the initial tilt) and arm movements at the LSD 16˚ position would not be reflected in the angle of SVV in the Control condition.

**Task during prolonged tilt.**　The participants performed one of the three tasks during prolonged tilt as follows: no-movement, static, or dynamic tasks. For the no-movement task, neither preparation nor action sounds were presented, and the participants were asked to maintain their tilted posture. For the static task, a preparation sound was first presented via the earphones, prompting the participants to switch the controller to their left hand and to

point to the front of the face using their right index finger with the right arm extended. The participants were instructed to maintain this posture until another preparation sound was presented. For the dynamic task, a preparation sound was first presented, and participants were asked to set the pointing posture as with the static tasks. Three seconds after the preparation sound, an action sound was presented every 3 s for a total of 10 times. The participants were asked to move their arm upward and then down parallel to their body's longitudinal axis once per action sound with their arm extended. The length of the arm movement was set from the height of the eye to the navel. Then, another preparation sound was presented, prompting them to hold the controller again.

Prior to the beginning of the experiment, the participants practiced each task sufficiently. The duration of each action condition (i.e., duration of prolonged tilt) was identical across all conditions (34 s). The order of the tasks was pseudorandomized across the participants.

**Data analysis.** The SVV angle was calculated as the deviation between the subjective vertical and actual gravitational vertical for each trial. The median of five trials was applied as the representative value for each task condition for each participant. To determine the extent of SVV shifts induced by prolonged tilt and arm movements, the $\Delta SVV$ values were calculated by subtracting the SVV angle before the task from that after it for the *during-tilt* session, and by subtracting the SVV angle in the Control condition from the SVV angle in each task condition (No-movement, Static, Dynamic) for the *post-tilt* session. The results of Shapiro-Wisk test showed that the SVV angles and $\Delta SVV$ values were normally distributed across participants in all conditions for the *during-tilt* session (all $p > 0.05$), but not in some conditions for the *post-tilt* session ($p < 0.05$). Therefore, parametric statistical analyses were applied to the dataset in the *during-tilt* session, while non-parametric analyses were applied to the dataset in the *post-tilt* sessions. Specifically, the $\Delta SVV$ in each task condition were compared using one-way analysis of variance (ANOVA; three task conditions [No-movement, Static, Dynamic]) with repeated measures for the *during-tilt* session and Friedman tests (three task conditions [No-movement, Static, Dynamic]) for the dataset in each final tilt position (0, LSD, or RSD 4˚) for the *post-tilt* session.

For the ANOVA, the degrees of freedom were corrected using Greenhouse-Geisser correction coefficient epsilon, and the *p*-value was recalculated if sphericity was violated with Mauchly's sphericity test. The significance level for all comparisons was set at $p < 0.05$. Bonferroni correction was used for post-hoc multiple comparisons. All statistical analyses were conducted using R software version 3.5.3 (R Core Development Team, Austria).

## Results

**Prolonged tilt effect.** We first checked whether prolonged tilt induced SVV shifts in the present experimental setup. Fig 3 shows the angular changes of the SVV in the No-movement condition for the *during-tilt* session and SVV angles in the Control and No-movement conditions for the *post-tilt* session. Positive and negative values correspond to rightward and leftward deviations, respectively. For the *during-tilt* session, a paired t-test revealed that the SVV angle after the no-movement task (1.3 ± 1.2˚) significantly shifted leftward compared to that before the task (-0.4 ± 1.3˚; $t_{11} = 2.58$; $p < 0.05$, Cohen's $d = 0.67$). For the *post-tilt* session, the median SVV angles (1st, 3rd quartiles) in the Control and No-movement conditions were -1.0˚ (-1.5, 0.7) and -2.7˚ (-6.0, -0.4) for the LSD 4˚ position, -1.5˚ (-2.5, 0.5), and -3˚ (-5.6, -0.9) for the 0˚ position, and -3.5˚ (-6.5, -1.5) and -4.0˚ (-8.6, -1.8) for the 4˚ RSD position. The SVV angles for the Control condition were not significantly different between the final tilt positions (Friedman test; main effect $\chi^2 = 2.07$; $p = 0.36$). The results of Wilcoxon signed ranked test showed that the SVV angle in the No-movement condition significantly shifted leftward

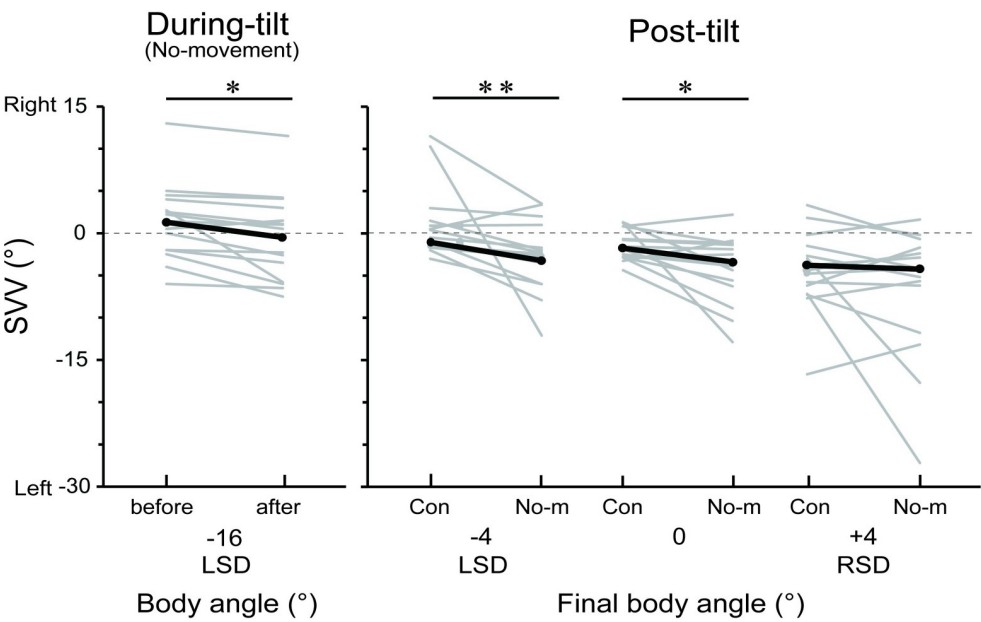

**Fig 3. The alteration of SVV angles during prolonged tilt without arm movements.** 'Con' and 'No-m' refer to the Control and No-movement conditions, respectively. Grey bars denote the individual median data, and black bars denote the group-mean (*during-tilt* session) or -median data (*post-tilt* session). $^{*}$: $p < 0.05$, $^{**}$: $p < 0.01$.

compared to the Control condition for LSD 4° ($p < 0.001$, effect size $r = 0.86$) and 0° positions ($p < 0.05$, effect size $r = 0.78$), but not for the RSD 4° position ($p = 0.35$, effect size $r = 0.28$).

**During-tilt session.** Fig 4A shows the group-mean $\Delta SVV$ value for each task condition. The results of one-way ANOVA revealed a significant main effect of task condition ($F_{2, 28} = 4.77$, $p < 0.05$, effect size $\eta^2 = 0.14$). Results of post-hoc Bonferroni tests showed that the $\Delta SVV$

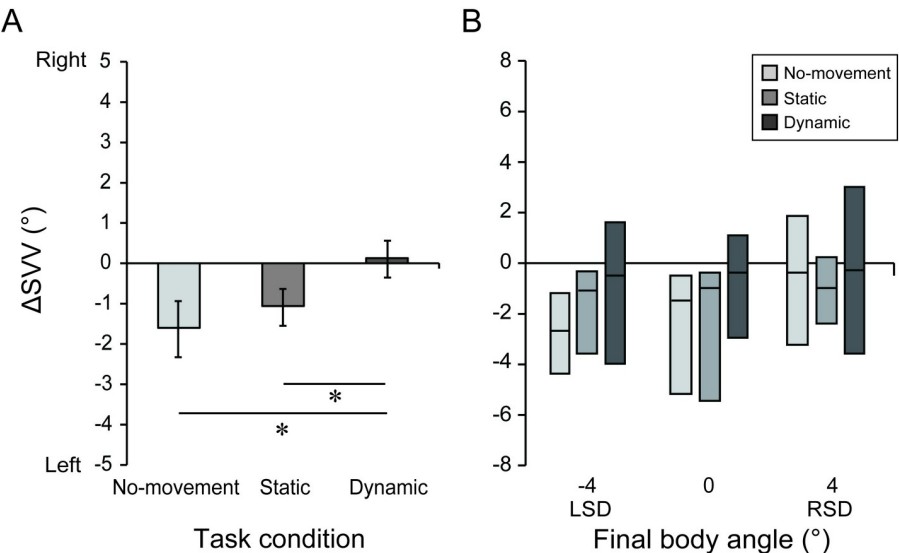

**Fig 4.** The group-mean $\Delta SVV$ values in the *during-tilt* (A) and *post-tilt* sessions (B). (A) Error bars denote standard error. (B) The horizontal line within each box and the lower and upper ends of each box represent the median and 1st and 3rd quartiles, respectively. $^{*}$: $p < 0.05$

in the No-movement (-1.6 ± 0.6˚) and Static conditions (-1.2 ± 0.4˚) were smaller (i.e., shifted leftward) than in the Dynamic condition (0.2 ± 0.4˚; vs No-movement, $p < 0.05$, effect size $r = 0.60$; vs Static, $p < 0.05$, effect size $r = 0.54$). No significant differences were noted between the Static and No-movement conditions ($p = 0.46$, effect size $r = 0.19$). These results indicate that the SVV shifts that occurred during prolonged tilt were attenuated when performing dynamic arm movements.

**Post-tilt session.** Fig 4B shows the group-median $\Delta SVV$ in each final tilt position for each action condition. The results of Friedman tests revealed a significant main effect of task condition for the LSD 4˚ position ($\chi^2 = 8.40$, $p < 0.05$), but not for 0˚ ($\chi^2 = 2.53$, $p = 0.28$) and RSD 4˚ ($\chi^2 = 0.85$, $p = 0.65$). For the LSD 4˚ position, however, the results of post hoc tests showed no significant differences in $\Delta SVV$ among different task conditions (No-movement vs Dynamic, $p = 0.13$, effect size $r = 0.58$; No-movement vs Static, $p = 0.15$, effect size $r = 0.52$; Static vs Dynamic, $p = 0.44$, effect size $r = 0.38$). These results indicate that the SVV shifts that occurred after prolonged tilt were not significantly influenced by either static or dynamic arm movements during prolonged tilt.

## Experiment 2

### Materials and methods

**Participants.**   Twelve right-handed healthy volunteers (8 men and 4 women, aged 22–26 years) participated in this experiment after providing written informed consent. Similar to Experiment 1, this experiment was approved by the Ethics Committee of the Graduate School of Human and Environmental Studies, Kyoto University, and was conducted in accordance with the Declaration of Helsinki (2013).

**Apparatus.**   As in Experiment 1, the participants sat on a tilting chair, and the head, trunk, and legs were firmly fastened to the seat. In this experiment, the velocities of the tilt table were different from those in Experiment 1. The velocity of the tilt-chair during the subjective postural vertical (SPV) task (see *Experimental Procedure* paragraph) was set at 1.0˚/s (initial acceleration: 0.52˚/s$^2$) to avoid the stimulation of semi-circular canals [34]. The velocity of the chair from upright to LSD 16˚ (before the SPV task) and from RSD 16˚ to the upright position (after the SPV task) was relatively fast (8.0˚/s). However, as a previous study [36] reported that the amplitude of post-rotatory nystagmus was small even after roll body tilt at a speed of 10˚/s, the contribution of the semicircular canal to SPV estimations was negligible.

The participants held a custom-made controller with a press button to indicate the position of the perceived body vertical. The roll-tilt angle of the chair was monitored using an accelerometer module (KXM52-1050, Kionix, USA) mounted at the center of the tilt table. The signals from the accelerometer and controller were recorded using a data acquisition system (Power Lab 16sp, AD Instruments, Australia). The sampling frequency was set to 100 Hz.

During the experiment, the participants wore an eye-mask and were provided with white noise via earphones so as not to provide visual or auditory cues from the environment. To prevent the participants from being fatigued, a rest period of approximately 10 min was inserted per 10 SPV trials.

**Experimental procedure.**   The blindfolded participants were first tilted to LSD 16˚. In this position, they were presented with one of four task conditions (No-movement, Static, Dynamic, Control). Under the former three conditions, as in Experiment 1, they were asked to perform each task according to the preparation and action sounds (see *Task during prolonged tilt*), and then they were tilted to RSD 16˚. While tilted from LSD 16˚ to RSD16˚, they were asked to press the bottom of the controller when they felt that their body was upright (SPV task). In the Control condition, they were tilted to RSD 16˚ immediately after arriving at LSD

16˚ (i.e., without prolonged tilt) and performed the SPV task. After each SPV task, the participants were tilted back to the upright position. As in Experiment 1, the duration of each task was 34 s.

All the participants performed 7 trials of the SPV task for each of the four task conditions, that is, a total of 28 trials. The order of presentation of the task conditions was pseudorandomized for each participant.

**Data analysis.** The SPV angle was calculated as the deviation between the subjective vertical and actual gravitational vertical for each trial. The median was used as the representative value of the SPV angles for each task condition for each participant. Similar to SVV, the extent of SPV shifts ($\Delta SPV$) induced by prolonged tilt and arm movements were quantified by subtracting the SPV angle in the Control condition from that in each task condition (No-movement, Static, and Dynamic). Since the SPV angles in each task condition were normally distributed across participants (Shapiro-Wisk tests, $p > 0.05$), a one-way ANOVA with repeated measures (three task conditions [No-movement, Static, Dynamic]) was conducted to compare $\Delta SPV$ values between task conditions. If sphericity was violated under Mauchly's sphericity test, the degree of freedom was corrected using the Greenhouse-Geisser correction coefficient epsilon, and the $p$-value was recalculated.

## Results

Fig 5A shows the mean SPV angles in the Control and No-movement conditions. The SPV angle significantly shifted leftward in the No-movement condition (-5.0 ± 0.9˚), compared to the Control condition (-0.8˚± 1.2˚; $t_{11}$ = 5.77, $p < 0.001$, Cohen's $d$ = 1.67). This result indicates significant SPV shifts induced by prolonged tilt.

Fig 5B shows the group-mean $\Delta SPV$ values for each task condition. The mean (±SE) $\Delta SPV$ were -4.3 ± 0.7˚ for the No-movement condition, -3.5 ± 0.8˚ for the Static condition, and -3.9 ± 0.8˚ for the Dynamic condition, respectively. The result of one-way ANOVA revealed a non-significant main effect of task condition ($F_{2, 22}$ = 2.54, $p$ = 0.10) with a small effect size ($\eta^2$ = 0.01). This result indicates that neither static nor dynamic arm movements significantly influenced the SPV shifts induced by prolonged tilt.

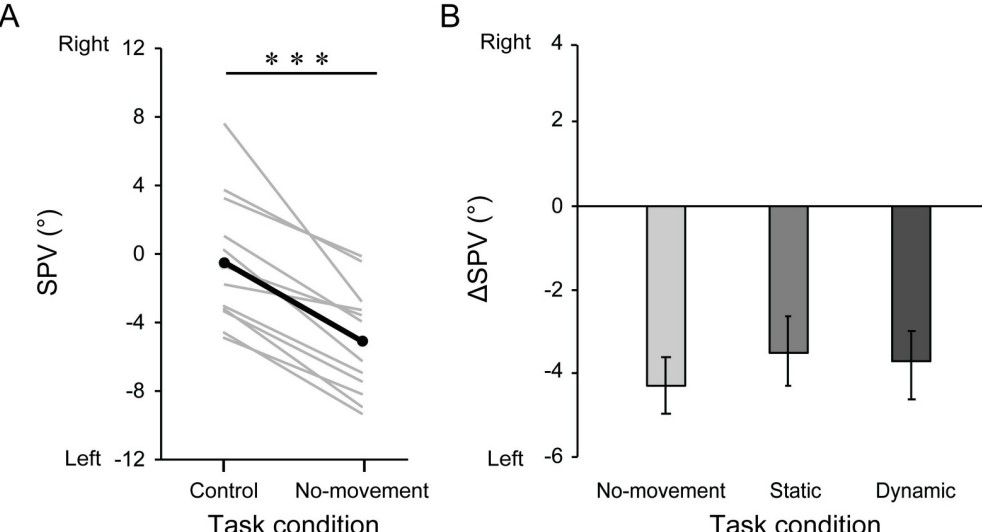

**Fig 5.** The SPV angles in the Control and No-movement conditions (A) and group-mean $\Delta SPV$ value in each task condition (B). (A) Gray and black lines represent the individual median and group-mean values, respectively. (B) Error bars represent standard errors. ***: $p < 0.001$.

## Discussion

The present study investigated how static or dynamic arm movements influenced changes in the SVV and SPV angles induced by prolonged tilt. In Experiment 1, we found that the performance of dynamic arm movements effectively attenuated the SVV shifts that occurred during prolonged tilt (*during-tilt* session), but not after prolonged tilt (*post-tilt* session). In Experiment 2, the SPV angles were not significantly affected by either static or dynamic arm movements.

Extending previous findings that the accuracy in spatial judgment at the tilted position was considerably improved by accompanying arm movements during judgment [32, 33], we hypothesized that active body movements could subsequently influence perceptual estimates of gravitational direction not involving body movements. Based on this hypothesis, we predicted that the perceptual distortion of the gravitational direction induced by prolonged tilt would decrease when active arm movements are performed in the tilted position. In support of our prediction, the results of the *during-tilt* session showed that the shifts of SVV toward the direction of prolonged tilt (Fig 3, *left panel*) were attenuated when the participants performed dynamic arm movements during prolonged tilt (Fig 4A). Prolonged tilt induces adaptive changes in the vestibular and body somatosensory systems [23, 28], leading to a decrease in the sensed angles of the head and/or body relative to gravity [29]. The performance of arm movements against gravity provides supplemental cues such as proprioceptive feedback or efferent copy [30] for estimating head and/or body orientation in space. The CNS would likely recalibrate the internal estimates of the gravitational direction based on these information, resulting in the stable perceptual judgment of visual vertical through prior knowledge/experience.

In contrast to dynamic arm movements, we observed no significant effects of static arm movements on SVV in the *during-tilt* session, even though the gravitational force on the arm, dependent on the body's tilt angle, is generated by both types of arm movements. The lack of an effect of static arm movements may imply that the dynamic property of arm movements is important for estimating the visual vertical. The gravitational force on the arm during arm movements can be perceived as a sense of heaviness based on the afferent information about muscle tension from the Golgi tendon organ (GTO) located at the muscle-tendon junction [37, 38]. Psychophysical studies have shown that estimation of the heaviness of an object with concurrent dynamic movements, such as lifting or wielding, is more accurate than estimation with static holding [39, 40]. In addition, a physiological study has demonstrated that when constant tension is persistently applied to a muscle, the firing rate and sensitivity of the GTO to the force gradually deteriorate [41]. These findings lead us to speculate that information about the gravitational force on the arm might be conveyed to the CNS more accurately while performing dynamic than static arm movements, leading to effective attenuation of the prolonged-induced SVV shifts.

In the *post-tilt* session, the significant SVV shifts after prolonged tilt were observed in the final tilt positions of LSD 4˚ and 0˚ (Fig 3, *right panel*). However, in contrast to the *during-tilt* session, dynamic arm movements did not significantly attenuate these SVV shifts (Fig 4B). One possible explanation for this difference between the sessions is the interval between the arm movement task and SVV adjustment. In the *during-tilt* session, the participants performed the SVV adjustments immediately after the arm movement task. On the other hand, in the *post-tilt* session, they performed the SVV adjustments after slowly tilting back toward each final tilt position; thus, the interval between dynamic arm movements and SVV adjustments was relatively long (at least 20s). The contribution of the additional cues derived from dynamic arm movements to visual vertical estimates likely diminished over time after the task, thereby

resulting in no clear effects of dynamic arm movements on SVV in the *post-tilt* session. In favor of this assumption, the attenuation effect of dynamic arm movements on SVV shifts appears to be greater at positions closer to the initial tilt position, where the interval between the action task and SVV adjustment was shorter.

The prolonged tilt-induced SPV shifts were not significantly influenced by either static or dynamic arm movements (Fig 5). Due to the relatively long time between the estimation of postural vertical and the arm movement task, as well as the *post-tilt* session, the lack of significant effect of arm movements on the SPV may also be attributed to the temporal decay of the arm movement effects. In contrast, a previous study showed that detection of self-body tilt was not improved even immediately after dynamic arm movements [42]. Given this, the present result in SPV may indicate that the arm movement-related gravitational cues are less utilized for the estimation of body orientation relative to gravity.

Although the results of the *during-tilt* session suggest the role of active body movements in the conscious perception of the gravitational direction, it remains unclear whether or how they influence the control of body orientation. Previous studies have shown an inconsistency between perceived and achieved body orientations when actively controlling body orientation [43, 44]. This suggests that dynamic body movements may have different effects on the perception of gravitational direction and control of body orientation. On the other hand, some recent studies have demonstrated the contribution of dynamic somatosensory cues to active postural control [45, 46]. Future research directly assessing the influence of dynamic arm movements on the achieved body orientation would be helpful for a better understanding of the mechanisms underlying the perception and control of body orientation in space.

Three limitations must be noted when interpreting our findings. First, the sample size was relatively small. In particular, at the LSD 4˚ position in the *post-tilt* session, no significant differences were noted between the No-movement condition and the Static or Dynamic conditions despite the large effect size ($r > 0.50$), which is likely a result of the small sample size. Therefore, our findings need to be confirmed by studies that include a larger sample size. Second, we used a static whole-tilt for the SVV assessment in which the gravitational and gravitoinertial force (GIF) vectors were the same; therefore, it remains unknown whether participants responded to either force vectors. Since the otolith system responds to both gravitational and inertial forces [28], the visual vertical estimation reflects a response to the GIF. However, a previous study has shown that the estimation of the earth-horizontal direction is differently influenced by whole-body tilt and body centrifugation, even though the GIF vector relative to the head was identical [33]. This implies that the gravitational force may specifically affect the perception of gravitational direction. To address this, further studies are needed to dissociate the gravitational and GIF vectors. Third, the arms were not restrained to the body during prolonged tilt. In such a situation, gravity would have pulled the arms to the side, providing a static cue for the perception of the gravitational direction even when the arm movements were not performed (i.e., No-movement condition). This methodological limitation may be partially responsible for the lack of a significant difference in the SVV angles between the No-movement and Static conditions.

## Conclusion

The present study shows that dynamic arm movements can attenuate the perceptual distortion of the visual vertical induced by prolonged tilt. This finding suggests that the supplementary information generated by dynamic body movements plays an important role in the perceptual estimates of gravitational direction as well as vestibular and body somatosensory signals. To provide a comprehensive understanding of the relationship between action and the perception

of the gravitational space, we need to further examine how performance in the estimation of the gravitational direction is influenced by the manipulation of temporal (e.g., arm movement velocity, interval between arm movements and perceptual tasks) and spatial properties (e.g., direction and angle of arm movements or body tilt).

## Author Contributions

**Conceptualization:** Keisuke Tani, Shinji Yamamoto, Yasushi Kodaka, Keisuke Kushiro.

**Data curation:** Keisuke Tani.

**Formal analysis:** Keisuke Tani.

**Funding acquisition:** Keisuke Tani, Keisuke Kushiro.

**Investigation:** Keisuke Tani.

**Methodology:** Keisuke Tani, Yasushi Kodaka, Keisuke Kushiro.

**Project administration:** Keisuke Tani, Shinji Yamamoto, Keisuke Kushiro.

**Resources:** Keisuke Tani, Keisuke Kushiro.

**Software:** Keisuke Tani.

**Supervision:** Keisuke Tani, Shinji Yamamoto, Yasushi Kodaka, Keisuke Kushiro.

**Validation:** Keisuke Tani.

**Visualization:** Keisuke Tani.

**Writing – original draft:** Keisuke Tani, Keisuke Kushiro.

**Writing – review & editing:** Keisuke Tani.

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
