## [Decision Letter · Decision Letter 0]

22 Jan 2021

PONE-D-20-40610

Dynamic arm movements attenuate perceptual distortion of visual vertical induced
during prolonged whole-body tilt

PLOS ONE

Dear Dr. Tani,

Thank you for submitting your manuscript to PLOS ONE. After careful consideration, we
feel that it has merit but does not fully meet PLOS ONE’s publication criteria as it
currently stands. Therefore, we invite you to submit a revised version of the
manuscript that addresses the points raised during the review process.

I have received comments from two knowledgeable Reviewers. Both have requested
significant changes but, in my judgment, the changes are tractable. I believe that a
solid revision will significantly enhance the value of your contribution.

Please submit your revised manuscript by Mar 05 2021 11:59PM. If you will need more
time than this to complete your revisions, please reply to this message or contact
the journal office at plosone@plos.org. When
you're ready to submit your revision, log on to https://www.editorialmanager.com/pone/ and select the 'Submissions
Needing Revision' folder to locate your manuscript file.

If you would like to make changes to your financial disclosure, please include your
updated statement in your cover letter. Guidelines for resubmitting your figure
files are available below the reviewer comments at the end of this letter.

We look forward to receiving your revised manuscript.

Kind regards,

Thomas A Stoffregen, PhD

Academic Editor

PLOS ONE

Journal Requirements:

Reviewers' comments:

Reviewer's Responses to Questions

**Comments to the Author**

1. Is the manuscript technically sound, and do the data support the conclusions?

Reviewer #1: Partly

Reviewer #2: Partly

2. Has the statistical analysis been performed
appropriately and rigorously? 

Reviewer #1: Yes

Reviewer #2: Yes

3. Have the authors made all data underlying the
findings in their manuscript fully available?

Reviewer #1: Yes

Reviewer #2: Yes

4. Is the manuscript presented in an intelligible
fashion and written in standard English?

Reviewer #1: Yes

Reviewer #2: Yes

5. Review Comments to the Author

Reviewer #1: Restrained subjects were passively tilted to a fixed angle in the
coronal plane. The subject’s task was to adjust a display (a line) to align with the
“gravitational vertical”. These subjective reports were made before and after arm
movements, and after the completion of passive body tilt. In Experiment 2, passive
body tilt was ongoing, and subjects were asked to indicate when they felt their body
to be aligned with “gravitational vertical”. The results replicated common findings
that the subjective vertical can be influenced by passive body tilt. The main
finding was that perceived orientation was influenced by arm movements.

The authors claim that “no experimental evidence” exists relating active body
movement to perception of “gravitational space”. This claim seems odd, given that
the authors have cited the work of Bringoux, who studied exactly this topic. In
addition, other studies have examined the role of active movement on perception of
orientation, in general, and the vertical in particular. Perhaps the closest, with
respect to the present study, is the work of Fouque et al. (1999), in which active
arm movements were related to whole body tilt in the perception of orientation. In
revising, I think the authors should explain how their hypotheses, design, results,
and interpretation differ from Fouque et al. In addition, please revise to indicate
that the present study provides information only about perception during passive
tilt, as contrasted with studies in which body tilt has been actively controlled
(e.g., Panic et al., 2015; Riccio et al., 1992). It would be specially helpful, in
the Discussion, to consider how future research might help us understand relations
between perceived orientation during passive versus active tilt. Achieved
orientation can differ from subjective orientation; moreover, outside the
laboratory, conscious awareness of orientation is uncommon, whereas (successful)
control of orientation is nearly continuous.

It is widely assumed that the body is controlled relative to the direction of
gravity, but this view is not universal. In fact, it has come under sustained
criticism, mainly because body movement is not constrained directly (or solely) by
the gravitational vector but, rather, by the sum of gravitational and inertial
forces—the gravitoinertial force vector (e.g., Stoffregen & Riccio, 1988). In
the present experiments, the gravitational and gravitorinertial force vectors were
the same, and so the results cannot tell us whether participants were responding to
one or the other. This limitation of the design should be noted in the Discussion.
Similarly, clinical data from stroke patients do not permit the scientist to know
which vector is detected.

The Introduction should be revised to state explicitly the hypotheses that were
tested in the study. What testable predictions did the authors make? Similarly, the
Discussion should be revised to re-state the predictions indicating, in each case,
whether each prediction was (or was not) confirmed. The pattern of confirmed (vs.
not confirmed) predictions should structure data interpretation.

Please revise so that the Results of Experiment 1 are presented before the Method of
Experiment 2. That is, completely present Experiment 1, and then completely present
Experiment 2.

Fouque, F., Bardy, B. G., Stoffregen, T. A., & Bootsma, R. B. (1999). Intermodal
perception of orientation during goal-directed action. Ecological Psychology, 11,
45-79.

Panic, H., Panic, A. S., DiZio, P. & Lackner, J. R. (2015). Direction of balance
and perception of the upright are perceptually dissociable. Journal of
Neurophysiology, 113, 3600-3609.

Riccio, G. E., Martin, E. J., & Stoffregen, T. A. (1992). The role of balance
dynamics in the active perception of orientation. Journal of Experimental
Psychology: Human Perception & Performance, 18, 624-644.

Stoffregen, T. A., & Riccio, G. E. (1988). An ecological theory of orientation
and the vestibular system. Psychological Review, 95, 3-14.

Reviewer #2: This interesting paper investigates dynamic arm movements as a new
variable in the long search for the sensory determinants of the subjective vertical.
The 'During Tilt' portion of Experiment 1 first re-demonstrates the known ability of
prolonged tilt to bias the subjective visual vertical (SVV) towards the tilt, and
then shows (as a new finding) that a series of dynamic arm movements during the tilt
reduces this bias. The 'Post Tilt' portion of Experiment 1 shows that this bias
reduction does not occur if the SVV is estimated after the participant is moved to a
different tilt angle. This may be due to the timing between the prolonged tilt and
the SVV estimation (as mentioned by the authors) but could also be due to the new
tilt angle 'overwriting' the participant's sense of orientation. Experiment 2 shows
that a subjective postural vertical estimation is not affected by the dynamic arm
movements in the same manner as the SVV estimation is. The sample sizes are on the
small side, as noted by the authors themselves, but the data analysis appears to be
well done. I recommend some revisions as follows:

Lines 46-92

The introduction is not as thorough as I would expect in an otherwise well-written
paper. Many of the references are grouped together with somewhat superficial
descriptions, such as in Lines 49-51. There are very few references from the past 10
years, despite considerable recent research from several labs on the use of dynamic
and movement cues on balance. The paper would benefit from a more substantive
introduction, which could then be used to deepen the discussion of the results.

Lines 104-106

Was there a reason not to secure the participants’ arms by some mechanism that could
be loosened/removed at the same time as the display frame was rotated to the left?
If the arms were left unsecured then they will necessarily be pulled to the side by
gravity when in a static roll, and this would provide an extra, potentially
confounding sensory cue. This may have been a reason why the no-movement and static
conditions did not significantly differ (Fig 4).

Lines 142-165, Lines 242-253

The term ‘trial’ appears to be used for two different types of event: a single
instance of the participant setting the white line for SVV (as in line 148), as well
as a set of ‘tilt, SVV, task, SVV’ (as in line 153). The explanation of the
procedure would be clarified if two different terms were used.

Line 159

What was the rationale for selecting to test at 4 degrees left, 0 degrees, and 4
degrees right? Would a larger tilt be expected to produce a larger effect?

Lines 160-162

It was not clear to me why the 'Post Tilt' experimental procedure ends with moving
the participant to 16 degrees right and then to the start position. These tilts
happen after the SVV estimations are made, so are they necessary? If they are
necessary for the 'Post Tilt' procedure, why are they not done at the end of the
'During Tilt' procedure?

Lines 378-382

The lack of effect of dynamic arm movements on SVV estimation in the 4 degree Right
position for the ‘Post Tilt’ procedure is explained as a result of a small sample
size. This may be true, but it may also represent an effect of always using a
prolonged left tilt at the start of the experiment. What would happen if the initial
tilt was to the right instead?

Figure 1:

This figure appears to be added to show how the display rotates in yaw away from the
participant. I found it confusing at first, because I was expecting an image showing
the roll rotation used in the experiment. Perhaps the figure could be updated to
show both of these features.

Figure 3:

Why is the SVV angle for the control in the +4 RSD group so different from the
controls in the +4 LSD and 0 degrees groups? I would think that all three groups
would have very similar values on the control.

Figure 3:

Why is the variability so much larger for the +4 RSD group, compared to the +4 LSD
and 0 degrees groups?

6. PLOS authors have the option to publish the peer
review history of their article (what does this mean?). If published, this will
include your full peer review and any attached files.

If you choose “no”, your identity will remain anonymous but your review may still be
made public.

**Do you want your identity to be public for this peer review?** For
information about this choice, including consent withdrawal, please see our
Privacy Policy.

Reviewer #1: No

Reviewer #2: No

---

## [Author Response · Author response to Decision Letter 0]

29 Mar 2021

Reviewer #1

Comment 1

Restrained subjects were passively tilted to a fixed angle in the coronal plane. The
subject’s task was to adjust a display (a line) to align with the “gravitational
vertical”. These subjective reports were made before and after arm movements, and
after the completion of passive body tilt. In Experiment 2, passive body tilt was
ongoing, and subjects were asked to indicate when they felt their body to be aligned
with “gravitational vertical”. The results replicated common findings that the
subjective vertical can be influenced by passive body tilt. The main finding was
that perceived orientation was influenced by arm movements.

Responses; 

We appreciate the constructive comments from the reviewers. We have carefully read
the comments and corrected our manuscript accordingly.

Comment 2

The authors claim that “no experimental evidence” exists relating active body
movement to perception of “gravitational space”. This claim seems odd, given that
the authors have cited the work of Bringoux, who studied exactly this topic. In
addition, other studies have examined the role of active movement on perception of
orientation, in general, and the vertical in particular. Perhaps the closest, with
respect to the present study, is the work of Fouque et al. (1999), in which active
arm movements were related to whole body tilt in the perception of orientation. In
revising, I think the authors should explain how their hypotheses, design, results,
and interpretation differ from Fouque et al. In addition, please revise to indicate
that the present study provides information only about perception during passive
tilt, as contrasted with studies in which body tilt has been actively controlled
(e.g., Panic et al., 2015; Riccio et al., 1992). It would be specially helpful, in
the Discussion, to consider how future research might help us understand relations
between perceived orientation during passive versus active tilt. Achieved
orientation can differ from subjective orientation; moreover, outside the
laboratory, conscious awareness of orientation is uncommon, whereas (successful)
control of orientation is nearly continuous.

Responses;

As pointed out by the reviewer, our description was inappropriate because other
papers (Bringoux et al. 2004, 2007; Scotto Di Cesare et al. 2014) have shown the
influence of arm movements on spatial judgments even though these results were not
positive. Thus, we have removed the description “no experimental evidence” from the
manuscript, and accordingly, we have modified the abstract. 

 Fouque et al. (1999) showed that concurrent arm movements improved accuracy in
judgments of the head-referenced eye level, proposing that action promotes spatial
judgment. However, this finding cannot tell us whether action also influences the
perceptual judgment of gravitational space, not involving body movements. As the CNS
would estimate the gravitational vertical by integrating not only sensory inputs but
also prior knowledge/experience (Clemens et al. 2011), active body movements may
also influence the subsequent perceptual estimates of the gravitational direction
via prior knowledge/experience. To assess this hypothesis, we evaluated whether or
how active arm movements during prolonged tilt subsequently influenced the
perceptual judgments of gravitational direction (SVV or SPV). The results of the
during-tilt session showed a significant attenuation of SVV shifts by dynamic arm
movements (Fig. 4), at least partially supporting our hypothesis. This result
extends Fouque et al.’s finding and suggests that action contributes to perceptual
estimates of gravitational space, even without accompanying body movements. We have
modified the text (Introduction and Discussion) to emphasize the differences between
our study and theirs. 

 As the reviewer indicates, the body orientation continuously achieved in our daily
life would not always be consistent with the conscious and perceived orientation of
vertical. Panic et al. (2015) showed that the dissociation of the direction of
balance (DOB) from the gravitational vertical influences the achieved body upright
during active tilt, but not perceived upright. Unfortunately, since our study used
only passive body tilt, it remains unknown whether or how prolonged tilt influences
the achieved orientation when actively controlling the body, and whether it is
modulated by arm movements. We have described the importance of directly assessing
the effects of active body movements on the control of body orientation in future
research.

Changes;

Abstract

Lines 24-28

Concurrent body movements have been shown to enhance the accuracy of spatial
judgment, but it remains unclear whether they also contribute to perceptual
estimates of gravitational space not involving body movements. To address this, we
evaluated the effects of static or dynamic arm movements during prolonged whole-body
tilt on the subsequent perceptual estimates of visual or postural vertical. 

Introduction

Lines 80-94

Previous studies have shown that the body tilt-induced errors in the judgment of the
head-referenced eye level considerably decreased when accompanied by arm movements
during judgment [32,33]. This finding suggests that active body movements can
improve the accuracy of spatial judgments, but it is unknown whether active body
movements also influence the perceptual estimates of gravitational space not
involving body movements. The CNS considers prior knowledge and experience as well
as sensory signals to estimate the gravitational vertical [7-9], allowing us to
hypothesize that additional cues generated by body movements may contribute to the
subsequent perceptual estimates of the gravitational direction via prior knowledge
and/or experience. To test this hypothesis, the present study evaluated whether
static or dynamic arm movements during prolonged tilt influenced the perceptual
judgments of visual vertical (Experiment 1) or postural vertical (Experiment 2). As
mentioned above, the internal estimates of the gravitational direction are distorted
during or after prolonged tilt, primarily due to sensory adaptation. We expected
that these distorted estimates might be corrected based on additional cues generated
by arm movements, resulting in the maintenance of SVV or SPV angles even after
prolonged tilt. 

Discussion

Lines 350-363

Extending previous findings that the accuracy in spatial judgment at the tilted
position was considerably improved by accompanying arm movements during judgment
[32,33], we hypothesized that active body movements could subsequently influence
perceptual estimates of gravitational direction not involving body movements. Based
on this hypothesis, we predicted that the perceptual distortion of the gravitational
direction induced by prolonged tilt would decrease when active arm movements are
performed in the tilt position. In support of our prediction, the results of the
during-tilt session showed that the shifts of SVV toward the direction of prolonged
tilt (Fig. 3, left panel) were attenuated when the participants performed dynamic
arm movements during prolonged tilt (Fig. 4A). Prolonged tilt induces adaptive
changes in the vestibular and body somatosensory systems [23,28], leading to a
decrease in the sensed angles of the head and/or body relative to gravity [29]. The
performance of arm movements against gravity provides supplemental cues such as
proprioceptive feedback or efferent copy [30] for estimating head and/or body
orientation in space. The CNS would likely recalibrate the internal estimates of the
gravitational direction based on these information, resulting in the stable
perceptual judgment of visual vertical through prior knowledge/experience.

Lines 399-408

Although the results of the during-tilt session suggest the role of active body
movements in the conscious perception of the gravitational direction, it remains
unclear whether or how they influence the control of body orientation. Previous
studies have shown an inconsistency between perceived and achieved body orientations
when actively controlling body orientation [43,44]. This suggests that dynamic body
movements may have different effects on the perception of gravitational direction
and control of body orientation. On the other hand, some recent studies have
demonstrated the contribution of dynamic somatosensory cues to active postural
control (Misiaszek et al. 2016, 2017). Future research directly assessing the
influence of dynamic arm movements on the achieved body orientation would be helpful
for a better understanding of the mechanisms underlying the perception and control
of body orientation in space.

Comment 3

It is widely assumed that the body is controlled relative to the direction of
gravity, but this view is not universal. In fact, it has come under sustained
criticism, mainly because body movement is not constrained directly (or solely) by
the gravitational vector but, rather, by the sum of gravitational and inertial
forces—the gravitoinertial force vector (e.g., Stoffregen & Riccio, 1988). In
the present experiments, the gravitational and gravitorinertial force vectors were
the same, and so the results cannot tell us whether participants were responding to
one or the other. This limitation of the design should be noted in the Discussion.
Similarly, clinical data from stroke patients do not permit the scientist to know
which vector is detected.

Responses;

As the reviewer indicates, we used only a static tilt for SVV adjustments, and thus
could not dissociate the gravitational and gravitoinertial force (GIF) vectors.
Therefore, we cannot conclude whether the perceptual judgments of the gravitational
vertical (SVV and SPV) resulted from the responses to either vector. This is a
limitation of the present study. As suggested by a number of studies (e.g.,
Stoffregen & Riccio, 1988), the otolith system responds to the GIF but not
solely to the gravitational force; the performance on SVV adjustments would be
mainly derived from GIF. On the other hand, a previous study has demonstrated that
the estimation of the earth-horizontal direction is influenced differently by
whole-body tilt and body centrifugation, even though the GIF vector relative to the
head was identical (Carriot et al., 2006). This finding suggests that gravitational
force may specifically affect the perception of the gravitational direction. To
address this, further studies are needed to dissociate the gravitational and GIF
vectors. We have added this description to the text as a limitation of this study.
Additionally, we have deleted the description of the relationship between postural
control and perception of gravitational vertical in stroke patients from the
introduction, since this assumption would not be definitive. 

Changes;

Discussion

Lines 413-421

Second, we used a static whole-tilt for the SVV assessment in which the gravitational
and gravitoinertial force (GIF) vectors were the same; therefore, it remains unknown
whether participants responded to either force vectors. Since the otolith system
responds to both gravitational and inertial forces [28], the visual vertical
estimation reflects a response to the GIF. However, a previous study has shown that
the estimation of the earth-horizontal direction is differently influenced by
whole-body tilt and body centrifugation, even though the GIF vector relative to the
head was identical [33]. This implies that the gravitational force may specifically
affect the perception of gravitational direction. To address this, further studies
are needed to dissociate the gravitational and GIF vectors.

Comment 4

The Introduction should be revised to state explicitly the hypotheses that were
tested in the study. What testable predictions did the authors make? Similarly, the
Discussion should be revised to re-state the predictions indicating, in each case,
whether each prediction was (or was not) confirmed. The pattern of confirmed (vs.
not confirmed) predictions should structure data interpretation.

Responses;

We agree with these comments. We had to explicitly state our hypotheses. As we have
responded above (please see Comment 2), our hypothesis was that active body
movements might also influence the subsequent perceptual estimates of the
gravitational direction without body movements. To test this hypothesis, we
evaluated whether static or dynamic arm movements during prolonged tilt influenced
the perceptual judgments of visual or postural vertical. We predicted that the
distorted estimate of the gravitational direction induced by prolonged tilt might be
corrected based on additional cues generated by arm movements. Supporting our
prediction, in the during-tilt session, the shifts of SVV were significantly
attenuated by dynamic arm movements. We have revised the introduction and discussion
sections for the readers’ better understanding of the aims and hypotheses of this
study and our interpretation of the results.

Changes;

Introduction

Lines 80-94

Previous studies have shown that the body tilt-induced errors in the judgment of the
head-referenced eye level considerably decreased when accompanied by arm movements
during judgment [32,33]. This finding suggests that active body movements can
improve the accuracy of spatial judgments, but it is unknown whether active body
movements also influence the perceptual estimates of gravitational space not
involving body movements. The CNS considers prior knowledge and experience as well
as sensory signals to estimate the gravitational vertical [7-9], allowing us to
hypothesize that additional cues generated by body movements may contribute to the
subsequent perceptual estimates of the gravitational direction via prior knowledge
and/or experience. To test this hypothesis, the present study evaluated whether
static or dynamic arm movements during prolonged tilt influenced the perceptual
judgments of visual vertical (Experiment 1) or postural vertical (Experiment 2). As
mentioned above, the internal estimates of the gravitational direction are distorted
during or after prolonged tilt, primarily due to sensory adaptation. We expected
that these distorted estimates might be corrected based on additional cues generated
by arm movements, resulting in the maintenance of SVV or SPV angles even after
prolonged tilt. 

Discussion

Lines 350-363

Extending previous findings that the accuracy in spatial judgment at the tilted
position was considerably improved by accompanying arm movements during judgment
[32,33], we hypothesized that active body movements could subsequently influence
perceptual estimates of gravitational direction not involving body movements. Based
on this hypothesis, we predicted that the perceptual distortion of the gravitational
direction induced by prolonged tilt would decrease when active arm movements are
performed in the tilt position. In support of our prediction, the results of the
during-tilt session showed that the shifts of SVV toward the direction of prolonged
tilt (Fig. 3, left panel) were attenuated when the participants performed dynamic
arm movements during prolonged tilt (Fig. 4A). Prolonged tilt induces adaptive
changes in the vestibular and body somatosensory systems [23,28], leading to a
decrease in the sensed angles of the head and/or body relative to gravity [29]. The
performance of arm movements against gravity provides supplemental cues such as
proprioceptive feedback or efferent copy [30] for estimating head and/or body
orientation in space. The CNS would likely recalibrate the internal estimates of the
gravitational direction based on these information, resulting in the stable
perceptual judgment of visual vertical through prior knowledge/experience.

Comment 5

Please revise so that the Results of Experiment 1 are presented before the Method of
Experiment 2. That is, completely present Experiment 1, and then completely present
Experiment 2.

Responses;

According to the reviewer’s suggestion, we have revised the text.

 

Reviewer #2

Comment 1

This interesting paper investigates dynamic arm movements as a new variable in the
long search for the sensory determinants of the subjective vertical. The 'During
Tilt' portion of Experiment 1 first re-demonstrates the known ability of prolonged
tilt to bias the subjective visual vertical (SVV) towards the tilt, and then shows
(as a new finding) that a series of dynamic arm movements during the tilt reduces
this bias. The 'Post Tilt' portion of Experiment 1 shows that this bias reduction
does not occur if the SVV is estimated after the participant is moved to a different
tilt angle. This may be due to the timing between the prolonged tilt and the SVV
estimation (as mentioned by the authors) but could also be due to the new tilt angle
'overwriting' the participant's sense of orientation. Experiment 2 shows that a
subjective postural vertical estimation is not affected by the dynamic arm movements
in the same manner as the SVV estimation is. The sample sizes are on the small side,
as noted by the authors themselves, but the data analysis appears to be well done. I
recommend some revisions as follows:

Responses;

We are deeply thankful for the reviewer’s helpful comments and advice. We have
responded to all the comments and modified the text accordingly. 

Comment 2

Lines 46-92

The introduction is not as thorough as I would expect in an otherwise well-written
paper. Many of the references are grouped together with somewhat superficial
descriptions, such as in Lines 49-51. There are very few references from the past 10
years, despite considerable recent research from several labs on the use of dynamic
and movement cues on balance. The paper would benefit from a more substantive
introduction, which could then be used to deepen the discussion of the results.

Responses;

Several recent studies have shown the involvement of dynamic cues in postural control
(e.g., Misiaszerk et al. 2016). However, the purpose of our study was to clarify the
mechanism underlying the perception of gravitational direction rather than postural
control. As the other reviewer points out (please see #Reviewer 1 Comment 2), the
achieved body direction when actively controlling the posture can differ from the
perceived orientation. Given this, we have avoided describing the findings of
postural control in the Introduction. On the other hand, as the reviewer indicated,
our description in the Introduction was too superficial. Therefore, in the new
version of the manuscript, we have described the underlying mechanism of the
perception of the gravitational direction in more detail based on recent findings,
and more explicitly stated our hypothesis in the Introduction. Moreover, we have
mentioned the possible effect of arm movements on the control of body orientation
based on the findings that dynamic somatosensory cues influence postural control in
the Discussion. 

Changes;

Introduction

Lines 46-52

Knowledge of the gravitational direction is fundamental to our action and perception
of the earth. The direction of gravity cannot be directly sensed; instead, it is
estimated in the brain based on several types of sensory information. Numerous
psychophysical studies have demonstrated the involvement of visual [1-3],
somatosensory [4-6], and vestibular sensory signals [3,7] in estimates of
gravitational direction. Moreover, recent studies using computational modeling have
shown that the central nervous system (CNS) weighs and combines these multisensory
signals with prior knowledge and experience about the earth-vertical direction in a
statistically optimal manner to resolve sensory ambiguity [7-9].

Lines 80-94

Previous studies have shown that the body tilt-induced errors in the judgment of the
head-referenced eye level considerably decreased when accompanied by arm movements
during judgment [32,33]. This finding suggests that active body movements can
improve the accuracy of spatial judgments, but it is unknown whether active body
movements also influence the perceptual estimates of gravitational space not
involving body movements. The CNS considers prior knowledge and experience as well
as sensory signals to estimate the gravitational vertical [7-9], allowing us to
hypothesize that additional cues generated by body movements may contribute to the
subsequent perceptual estimates of the gravitational direction via prior knowledge
and/or experience. To test this hypothesis, the present study evaluated whether
static or dynamic arm movements during prolonged tilt influenced the perceptual
judgments of visual vertical (Experiment 1) or postural vertical (Experiment 2). As
mentioned above, the internal estimates of the gravitational direction are distorted
during or after prolonged tilt, primarily due to sensory adaptation. We expected
that these distorted estimates might be corrected based on additional cues generated
by arm movements, resulting in the maintenance of SVV or SPV angles even after
prolonged tilt. 

Discussion

Lines 399-408

Although the results of the during-tilt session suggest the role of active body
movements in the conscious perception of the gravitational direction, it remains
unclear whether or how they influence the control of body orientation. Previous
studies have shown an inconsistency between perceived and achieved body orientations
when actively controlling body orientation [43,44]. This suggests that dynamic body
movements may have different effects on the perception of gravitational direction
and control of body orientation. On the other hand, some recent studies have
demonstrated the contribution of dynamic somatosensory cues to active postural
control (Misiaszek et al. 2016, 2017). Future research directly assessing the
influence of dynamic arm movements on the achieved body orientation would be helpful
for a better understanding of the mechanisms underlying the perception and control
of body orientation in space.

Comment 3

Lines 104-106

Was there a reason not to secure the participants’ arms by some mechanism that could
be loosened/removed at the same time as the display frame was rotated to the left?
If the arms were left unsecured then they will necessarily be pulled to the side by
gravity when in a static roll, and this would provide an extra, potentially
confounding sensory cue. This may have been a reason why the no-movement and static
conditions did not significantly differ (Fig 4).

Responses;

Although the participants’ arms should have been restrained, we could not do this
methodologically. As indicated by the reviewer, gravity would pull the arms to the
side during prolonged tilt, which might provide a cue for the estimation of the
gravitational direction. This is another limitation of the present study. We have
described this as a limitation in the Discussion section. 

Changes;

Discussion

lines 421-425

Third, the arms were not restrained to the body during prolonged tilt. In such a
situation, gravity would have pulled the arms to the side, providing a static cue
for the perception of the gravitational direction even when the arm movements were
not performed (i.e., No-movement condition). This methodological limitation may be
partially responsible for the lack of a significant difference in the SVV angles
between the No-movement and Static conditions.

Comment 3

Lines 142-165, Lines 242-253

The term ‘trial’ appears to be used for two different types of event: a single
instance of the participant setting the white line for SVV (as in line 148), as well
as a set of ‘tilt, SVV, task, SVV’ (as in line 153). The explanation of the
procedure would be clarified if two different terms were used.

Reponses; 

We apologize for the inappropriate writing. To avoid confusion, we have applied the
trial “trial” for a single SVV adjustment, and “sequence of experiment trials” for a
set of trials in each condition in the new version of the manuscript.

Changes;

Materials and Methods

Lines 148

Figure 2A shows a sequence of experimental trials during the during-tilt session.

Lines 153-154

The participants performed five trials of the SVV adjustment within 40 seconds.

Lines 156-160

After the display portion was returned to the initial position (i.e., in front of the
participant’s face), the shutter opened and the participants were asked to perform
the SVV adjustments for five trials again. Each participant performed this sequence
of experimental trials for each task condition, that is, 30 trials (three task
conditions [No-movement, Static, Dynamic tasks] × 2 phases [before, after task] × 5
SVV adjustments) in total. 

Lines 167-168

The shutter opened and the participants were asked to repeat the SVV adjustments for
5 trials.

Lines 171-173

Each participant performed this sequence of trials for each task condition in each
final tilt position, i.e. 45 trials [3 task conditions (No-movement, Static, Dynamic
tasks) × 3 final tilt positions (0°, ±4°) × 5 SVV adjustments] in total.

Comment 4

Line 159

What was the rationale for selecting to test at 4 degrees left, 0 degrees, and 4
degrees right? Would a larger tilt be expected to produce a larger effect?

Responses;

We apologize for this inadequate explanation. Our interest in the post-tilt session
was to evaluate how arm movements during prolonged tilt attenuated the after-effect
of prolonged tilt on SVV angles near upright. Based on a previous finding showing
that approximately 4°is the threshold for the detection of body tilt relative to
gravity (Bringoux et al. 2002, Neuropsychologia), we assumed that participants could
recognize a body tilt at 4°, and used these angles. We have added the reason for the
application of such small tilt angles to the text. 

 The results of the post-tilt session showed that the effect of dynamic arm movements
on the SVV shifts tended to be larger (i.e., SVV shifts were more strongly
attenuated) at a position closer to the initial tilt position (Fig. 5), probably due
to the duration between the SVV and action tasks (as described in line 380-388).
Given this, we expect to observe a greater effect of dynamic arm movements at a
position (e.g., LSD 8°) closer to the prolonged tilt position. 

Changes;

Materials and Methods

Lines 145-146

These angles were determined based on the fact that 4° is the threshold for the
detection of body tilt in the roll plane [35].

Comment 5

Lines 160-162

It was not clear to me why the 'Post Tilt' experimental procedure ends with moving
the participant to 16 degrees right and then to the start position. These tilts
happen after the SVV estimations are made, so are they necessary? If they are
necessary for the 'Post Tilt' procedure, why are they not done at the end of the
'During Tilt' procedure?

Responses; 

In the post-tilt session, we set three final tilt positions. If the participants were
tilted back to the upright position directly from each position after the SVV task,
the tilt motion would provide a cue about the final tilt position, which could
affect the performance in subsequent trials. To prevent this as much as possible,
the participants were returned to the upright position via the RSD 16° position. In
the during-tilt session, only one body tilt angle was used; therefore, we did not
specifically consider this feedback. If multiple tilt directions and angles were
used in the during-tilt session, we inserted a tilt position before returning to
upright, as in the post-tilt session. To convey this point more clearly, we have
corrected the description of the reason for using RSD 16°in the text.

Changes;

Materials and Methods

Lines 168-170

After completing the task, the body was returned to the upright position via the RSD
16° position to avoid providing feedback about the final tilt position that could
influence the subsequent performance on the SVV adjustment.

Comment 6

Lines 378-382

The lack of effect of dynamic arm movements on SVV estimation in the 4 degree Right
position for the ‘Post Tilt’ procedure is explained as a result of a small sample
size. This may be true, but it may also represent an effect of always using a
prolonged left tilt at the start of the experiment. What would happen if the initial
tilt was to the right instead?

Responses;

The results in the post-tilt session show a lack of significant effect of dynamic arm
movements, not for only RSD 4°, but also for LSD 4° and 0° (please see Fig. 5).
However, the effect of arm movements tended to be smaller for the RSD 4° position
than for the other positions (although not statistically significant). As indicated
by the reviewer, the tilt direction used for prolonged tilt would have contributed
to this. The involvement of the additional information generated by arm movements in
the internal estimates of the gravitational direction presumably decays with time
(as described in lines 380-388). Because the time duration between the SVV and
action tasks was longer in RSD 4° than in LSD 4°, the effect of dynamic arm
movements may have been less observed in RSD 4° position. We speculate that if the
side of prolonged tilt was right, the effect of dynamic arm movements would be
larger in RSD 4° than in LSD 4°. We have not specifically stated this because it is
speculative. Instead, we have mentioned that the dependency of the effect of dynamic
arm movements on final tilt positions may be due to the duration between the SVV and
action tasks. In conclusion, we have stated the necessity of evaluating the effect
of dynamic arm movements using different tilt directions and angles in the
future.

Changes;

Discussion

Lines 388-390

In favor of this assumption, the attenuation effect of dynamic arm movements on SVV
shifts appears to be greater at positions closer to the initial tilt position, where
the interval between the action task and SVV adjustment was shorter.

Conclusion

Lines 433-438

To provide a comprehensive understanding of the relationship between action and the
perception of the gravitational space, we need to further examine how performance in
the estimation of the gravitational direction is influenced by the manipulation of
temporal (e.g., arm movement velocity, interval between arm movements, and
perceptual task) and spatial properties (e.g., direction and angle of arm movements
or body tilt).

Comment 7

Figure 1:

This figure appears to be added to show how the display rotates in yaw away from the
participant. I found it confusing at first, because I was expecting an image showing
the roll rotation used in the experiment. Perhaps the figure could be updated to
show both of these features.

Responses;

We apologize for the confusing depictions. We have modified the figure to better
depict our experimental setup and added a sentence to the figure caption. 

Changes;

Figure caption (Fig.1)

This figure illustrates a situation in which the participants were tilted leftward.
The display portion was rotated in yaw, as denoted by a gray arrow.

Comment 8

Figure 3:

Why is the SVV angle for the control in the +4 RSD group so different from the
controls in the +4 LSD and 0 degrees groups? I would think that all three groups
would have very similar values on the control.

Responses;

As pointed out by the reviewer, SVV angles for the Control condition appear to be
larger at RSD 4° than at LSD 4° and 0°. However, since our additional analysis
showed no significant difference in SVV angles for the Control condition between
these positions, we cannot conclude the larger SVV errors specifically for the RSD
4° position. Although we cannot exactly explain the reason for the tendency (not
statistically significant), this may reflect the structural and functional
properties of otolith organs given that the CNS would weigh more heavily on otolith
signals for the SVV adjustments near upright than somatosensory signals (Clemens et
al. 2011). Since this assumption is speculative, we have not specifically described
this in the text. Instead, we have added the statistical results showing no
significant difference in the angle of SVV for the control condition between the
final tilt positions. 

Changes;

Discussion

Lines 232-237

For the post-tilt session, the median SVV angles (1st, 3rd quartiles) in the Control
and No-movement conditions were -1.0° (-1.5, 0.7) and -2.7° (-6.0, -0.4) for the LSD
4° position, -1.5° (-2.5, 0.5), and -3° (-5.6, -0.9) for the 0° position, and -3.5°
(-6.5, -1.5) and -4.0° (-8.6, -1.8) for the 4° RSD position. The SVV angles for the
Control condition were not significantly different between the final tilt positions
(Friedman test; main effect χ2 = 2.07; p = 0.36). 

Comment 9

Figure 3:

Why is the variability so much larger for the +4 RSD group, compared to the +4 LSD
and 0 degrees groups?

Responses;

The inter-individual variability in the SVV angles appeared to be larger for the RSD
4°position than for the other positions. Unfortunately, as with Comment 7, we cannot
explain the reason for this based on our data. The characteristics of the otolith
function may also contribute to this. We have not specifically stated this in the
text because it is too speculative. We need to address this issue in future
studies.

---

## [Decision Letter · Decision Letter 1]

15 Apr 2021

Dynamic arm movements attenuate the perceptual distortion of visual vertical induced
during prolonged whole-body tilt

PONE-D-20-40610R1

Dear Dr. Tani,

We’re pleased to inform you that your manuscript has been judged scientifically
suitable for publication and will be formally accepted for publication once it meets
all outstanding technical requirements.

Kind regards,

Thomas A Stoffregen, PhD

Academic Editor

PLOS ONE

Additional Editor Comments (optional):

Reviewers' comments:

Reviewer's Responses to Questions

**Comments to the Author**

1. If the authors have adequately addressed your comments raised in a previous round
of review and you feel that this manuscript is now acceptable for publication, you
may indicate that here to bypass the “Comments to the Author” section, enter your
conflict of interest statement in the “Confidential to Editor” section, and submit
your "Accept" recommendation.

Reviewer #1: All comments have been addressed

Reviewer #2: All comments have been addressed

2. Is the manuscript technically sound, and do the data
support the conclusions?

Reviewer #1: Yes

Reviewer #2: Yes

3. Has the statistical analysis been performed
appropriately and rigorously? 

Reviewer #1: Yes

Reviewer #2: Yes

4. Have the authors made all data underlying the
findings in their manuscript fully available?

Reviewer #1: Yes

Reviewer #2: Yes

5. Is the manuscript presented in an intelligible
fashion and written in standard English?

Reviewer #1: Yes

Reviewer #2: Yes

6. Review Comments to the Author

Reviewer #1: (No Response)

Reviewer #2: (No Response)

7. PLOS authors have the option to publish the peer
review history of their article (what does this mean?). If published, this will
include your full peer review and any attached files.

If you choose “no”, your identity will remain anonymous but your review may still be
made public.

**Do you want your identity to be public for this peer review?** For
information about this choice, including consent withdrawal, please see our
Privacy Policy.

Reviewer #1: No

Reviewer #2: No

---

## [Editor Report · Acceptance letter]

22 Apr 2021

PONE-D-20-40610R1 

Dynamic arm movements attenuate the perceptual distortion of visual vertical induced
during prolonged whole-body tilt 

Dear Dr. Tani:

I'm pleased to inform you that your manuscript has been deemed suitable for
publication in PLOS ONE. Congratulations! Your manuscript is now with our production
department. 

Kind regards, 

on behalf of

Dr. Thomas A Stoffregen 

Academic Editor

PLOS ONE